# Are Viral Infections Key Inducers of Autoimmune Diseases? Focus on Epstein–Barr Virus

**DOI:** 10.3390/v14091900

**Published:** 2022-08-27

**Authors:** Masami Takei, Noboru Kitamura, Yosuke Nagasawa, Hiroshi Tsuzuki, Mitsuhiro Iwata, Yasuko Nagatsuka, Hideki Nakamura, Kenichi Imai, Shigeyoshi Fujiwara

**Affiliations:** 1Division of Hematology and Rheumatology, Department of Medicine, Nihon University School of Medicine, 30-1 Oyaguchi-Kamimachi, Itabashi-ku, Tokyo 173-8610, Japan; 2Division of Immunology and Pathobiology, Department of Microbiology, Dental Research Center, Nihon University School of Dentistry, 1-8-13 Kanda-Surugadai, Chiyoda-ku, Tokyo 101-8310, Japan; 3Department of Allergy and Clinical Immunology, National Research Institute for Child Health and Development, 2-10-1 Okura, Setagaya-ku, Tokyo 157-8535, Japan

**Keywords:** autoimmune disease, virus, Epstein–Barr virus, rheumatoid arthritis, signaling lymphocytic activation molecule-associated protein (*SAP*)/*SH2D1A*

## Abstract

It is generally accepted that certain viral infections can trigger the development of autoimmune diseases. However, the exact mechanisms by which these viruses induce autoimmunity are still not understood. In this review, we first describe hypothetical mechanisms by which viruses induce some representative autoimmune diseases. Then, we focus on Epstein–Barr virus (EBV) and discuss its role in the pathogenesis of rheumatoid arthritis (RA). The discussion is mainly based on our own previous findings that (A) EBV DNA and its products *EBV-encoded small RNA (EBER)* and latent membrane protein 1 (LMP1) are present in the synovial lesions of RA, (B) mRNA expression of the signaling lymphocytic activation molecule-associated protein (*SAP*)/*SH2D1A* gene that plays a critical role in cellular immune responses to EBV is reduced in the peripheral T cells of patients with RA, and (C) EBV infection of mice reconstituted with human immune system components (humanized mice) induced erosive arthritis that is pathologically similar to RA. Additionally, environmental factors may contribute to EBV reactivation as follows: *Porphyromonas gingivalis* peptidylarginine deiminase (PAD), an enzyme required for citrullination, engenders antigens leading to the production of citrullinated peptides both in the gingiva and synovium. Anti-citrullinated peptides autoantibody is an important marker for diagnosis and disease activity of RA. These findings, as well as various results obtained by other researchers, strongly suggest that EBV is directly involved in the pathogenesis of RA, a typical autoimmune disease.

## 1. Introduction

Viruses have various mechanisms to escape immune defense mechanisms of the host and establish persistent infection. For example, cytomegalovirus (CMV) inhibits presentation of viral antigens via HLA class I on the cell surface by dislocating HLA class I heavy chains from the endoplasmic reticulum to the cytosol [1]. The Nef protein encoded by human immunodeficiency virus (HIV) suppresses HLA class I expression and lets the virus escape from CTL recognition [2,3]. In addition, a cell surface TNF receptor family protein induced by the Nef protein suppresses activation of T cells and their apoptosis inducing function [4]. Recently, Epstein–Barr virus (EBV) microRNAs were reported to reduce the ability of CD8+ cytotoxic T cells to recognize antigen peptides by targeting IL-12, resulting in immune evasion [5]. Although the precise mechanisms of virus-induced autoimmunity have not been elucidated, recent evidence suggests that microRNA could play important roles (reviewed in [5,6,7]).

The development of autoimmune diseases following the establishment of persistent viral infections is often experienced in actual clinical settings and it is generally accepted that certain viral infections can trigger the development of autoimmune diseases. Typically, parvovirus B19, hepatitis virus B and C, human T-cell leukemia virus type 1, rubella virus, HIV, and CMV have been implicated in the development of autoimmune diseases (Table 1) [6,7,8,9,10,11,12,13,14,15,16,17,18,19,20,21,22,23,24,25,26,27]. Recent strong epidemiological evidence linking EBV and MS, as well as the association of EBV with other rheumatic diseases, has been reviewed [9,10]. Arthritis, uveitis, erythema nodosum, typical symptoms of rheumatic diseases, and representative symptoms of autoimmune diseases are often experienced in daily medical practice following viral infections (Table 2) [28,29,30,31,32,33,34,35].

The breakdown of immune tolerance induces autoimmunity and causes various autoimmune diseases. Although the main mechanisms of immune responses have been largely elucidated, the mechanisms of autoimmunity are still largely unknown. In this article, we discuss the hypothetical mechanisms of autoimmunity induced by viral infections, including our own hypothesis on the mechanism underlying EBV-induced development of rheumatoid arthritis.

## 2. Mechanism of Immune Tolerance Disruption by Viruses

Various theories have been presented regarding the mechanism by which viruses disrupt immune tolerance and induce autoantibody production [38]. Representative theoretical mechanisms by which viruses disrupt immune tolerance are listed in Table 3. Molecular mimicry is a typical example of such mechanisms. Autoimmune diabetes is clinically known to be associated with cytomegalovirus [37] and Coxsackie B virus. The amino acid sequence of glutamic acid decarboxylase (GAD65) expressed in pancreatic beta cells, a target molecule of autoantibodies in type I diabetes, shows homology with that of the P2-C protein encoded by Coxsackie B virus. It was also reported that a mouse model of autoimmune diabetes (NOD mouse) that harbored an anti-GAD65 antibody also had an autoantibody against insulin and other cellular proteins (epitope spreading) [39]. These findings suggest that the virus may be critically involved in the early stages of pathogenesis in autoimmune diabetes. In a mouse infection model, Coxsackie B virus was shown to release self-antigens from the myocardial tissue into the blood stream; these self-antigens activate the innate immune system and induce autoreactive T cells (bystander activation [40].

Type I (α and β) interferons (IFNs) play important roles in protection against viral infections. In daily practice, autoimmune reactions are found to be triggered by IFNs used for the treatment of viral hepatitis. IFNs enhance HLA class II expression. Molecules that have lost intramolecular S-S bonds (misfolds) are not rapidly degraded in the endoplasmic reticulum and are presented on the cell surface by HLA class II (neo-self). The immune response to neo-self has been reported to cause autoimmune phenomena [41]. Type I IFNs are also reported to reduce regulatory T-cell function. IFNs are thus key cytokines in the pathogenesis of autoimmunity [42].

Qingyong et al. proposed a new mechanism for the virus-induced development of autoimmune disease. They identified CD8+ T cells with dual T-cell receptors that recognize a self-antigen and a virus-specific protein [43]

In the following sections, we describe the pathophysiology of intractable autoimmune diseases associated with EBV infection and present our hypothesis on the mechanisms of their pathogenesis.

## 3. EBV-Associated Diseases

Recently, human herpesvirus (HHV)-related illnesses have been increased by the increasing use of immunosuppressive drugs for rheumatic diseases, organ transplantation, cancer, and HIV infection in adults. EBV (HHV4) is known to cause infectious mononucleosis and mainly infects B lymphocytes, causing Burkitt’s lymphoma and nasopharyngeal carcinoma. Many other EBV-related tumors have been reported, including Hodgkin’s disease, B-lymphoproliferative disease, and EBV-related gastric cancer. More than 90% of Japanese adults are infected with EBV. EBV is an important pathogen after solid organ transplantation. Most post-transplant viral infections occur within 180 days after transplantation. In the interim period of 30–180 days, virus reactivation from the donor organs and blood occurs. Reactivations of EBV, CMV, and varicella-zoster virus (VZV) peak once at this time. By the 180th day and later, CMV infection peaks again and, during this time, post-transplant lymphoproliferative disorder (PTLD) associated with EBV is observed. Some cases show late reactivation years after transplantation. Primary immunodeficiencies are also known to cause unique conditions associated with EBV-positive lymphoproliferation and/or hemophagocytic lymphohistiocytosis. Among them, X-linked lymphoproliferative disease type 1 (XLP-1) is known for its unique vulnerability to EBV. In addition to B cells, EBV occasionally infects T cells or natural killer (NK) cells and rarely induces T/NK-cell lymphoproliferative diseases such as chronic active EBV infection. Furthermore, EBV is considered to be involved in intractable autoimmune diseases such as IgA nephropathy, rheumatoid arthritis (RA), systemic lupus erythematosus (SLE), and multiple sclerosis.

## 4. EBV and Autoimmune Diseases

Patients with a number of autoimmune diseases show higher serum EBV antibody titers, higher blood EBV DNA load, and reduced EBV-specific cytotoxic T-cell response, as compared with controls. Autoantigen proteins involved in the pathogenesis of certain autoimmune diseases are molecularly mimicked by EBV-encoded proteins. Hypothetical mechanisms of the EBV-induced development of these autoimmune diseases were reviewed recently [36]. Very recently, a striking finding has been reported that may provide a unified theory explaining the pathogenesis of a wide spectrum of EBV-associated autoimmune diseases. Harley et al. have developed a program to quantitatively analyze the degree of correspondence between disease risk loci on the human genome and the binding sites of cellular and viral transcription factors. Many of the risk loci in SLE, MS, and RA have been shown to be occupied by the EBV-encoded transcription factor EBNA (EBV nuclear antigen)-2 [44]. At some risk loci, EBNA-2 is selectively bound to high-risk alleles and promotes their transcription. In other words, transcriptional promotion of a specific gene by EBNA-2 is suggested to be a trigger for the development of some autoimmune diseases. This study may thus be groundbreaking as it clarifies the interaction between genetic factors (risk loci) and environmental factors (EBV) in autoimmune diseases.

In the next section, using RA as an example, we try to explain the relationship between autoimmune diseases and viruses based on our own studies.

## 5. Involvement of EBV in RA Development

In addition to the findings described in the previous section that associate several autoimmune diseases with EBV, there are more specific finding suggesting the involvement of the virus in RA pathogenesis. Tan et al. discovered an antibody against the nuclear component of EBV-infected cells, RA nuclear antigen antibody (RANA), in the peripheral blood of RA patients [45]. The major epitope recognized by RANA is the glycine-alanine repeat structure of EBNA-1 [46,47,48]. A protein with a partial structural similarity to EBNA-1 has been identified in RA synovitis lesions [46,49]. Molecular mimicry (molecular homology) may thus play an important role in RA pathogenesis. It is also reported that the EBV-encoded glycoprotein gp110 contains the amino acid sequence QKRAA that is shared by alleles of HLA-DR4 and DR1 that are associated with a high risk of RA (shared epitope) [50]. Both humoral and T-cell-mediated cross-reactivities between EBV gp110 and HLA-DR4 have been described, although the exact mechanism of their involvement in RA pathogenesis is not clear. Anti-cyclic citrullinated protein (CCP) antibodies are autoantibodies that are highly specific to RA. It has been reported that the EBNA1 protein with its arginine residues of the glycine-arginine repeat structure deiminated is recognized by these anti-CCP antibodies [51], suggesting that EBV infection may trigger the production of anti-CCP antibodies. As the presence of autoantibody-producing B cells infected with EBV was demonstrated in ectopic lymphoid follicles in lesions of several other autoimmune diseases, EBV-infected autoantibody-producing plasmablasts have also been demonstrated in synovial ectopic lymphoid follicles of RA lesions [52].

As mentioned above, the association of RA and EBV had been examined from various viewpoints. However, for many years, the evidence for direct EBV infection to synovial cells of RA lesions had not been obtained [53,54], although the establishment of an EBV-positive fibroblast cell line from the RA synovium had suggested a certain role of the virus in the generation of arthritis lesion [55].

In 1997, we showed for the first time the presence of EBV in the synovium of RA lesions using polymerase chain reaction (PCR) and in situ hybridization [56]. The frequency of *EBER (**EBV-encoded small RNA)-1* positive cells tended to be higher in histological subtypes of RA with high higher degrees of lymphocyte and plasma cell infiltration. LMP-1 (latent membrane protein 1) expression was also observed in *EBER*-positive synovium. Expression of CD21 and EBNA-2 was not found at the same site. In RA synovial lesions, *EBER-1* and LMP-1 tended to be localized in apical regions of villous proliferation. These results were subsequently confirmed by a number of similar independent works [57,58,59]. Takeda et al. reported the expression of *EBER-1* and EBV DNA in RA synovial cells by in situ hybridization [58]. Saal et al. used RT-PCR and PCR to show that EBV DNA and *EBER-1* were detected at a high rate in RA lesions of patients with the HLA-DRB1 * 0401, 0404, 0405, or 0408 allele, and that the frequency of EBV positivity in RA lesions was 41 times higher than that in the normal controls [60]. Furthermore, Mahraein et al. reported the expression of *EBER-1/2* and LMP-1 in synovial lining cells [59]. In their report, 5 of 29 RA cases (17.2%) possessed infected cells in synovial lining cells and in mesenchymal cells, of which 3 cases showed localization mainly to surface cells. In addition, Takeda et al. reported that *BZLF-1* was expressed in only a small number of synovial mesenchymal cells from 1 of 29 patients with RA and from 1 of 26 patients with psoriatic arthritis, indicating that the main mode of EBV infection in the synovium was latent [58]. Brousset et al. also reported that the expression of EBV lytic cycle genes such as *BZLF-1* was not detected in RA lesions [61]. Meanwhile, Brousset et al. and Niedobiteke et al. detected EBV in lymphocytes in the RA synovium but not in synovial lining cells [53,62]. Mahraein et al. detected EBV-infected synovial lining cells in 1/6 and 1/26 cases of psoriatic arthritis and reactive arthritis, respectively, suggesting that EBV may be present in the lesions of a small percentage of patients with arthritis other than RA [59]. It can thus be summarized that a majority of works thus far published agree with the presence of EBV in synovial lining cells, although there have been a few discrepant results. The reduced EBV-specific CTL activity observed in RA patients [63] may explain why the presence of EBV-infected cells is tolerated in the RA synovium. LMP-1 has various functions; it increases the activity of the nuclear protein transcription activator NF-κB (nuclear factor-κB), increases the production of Bcl-2 and A20, and inhibits apoptosis [64,65]. LMP-1 is known to play a critical role in the growth-transformation of B cells by EBV and has the ability to convert human keratinocytes and rat fibroblasts into malignant cells, suggesting that it may also be involved in the proliferation of synovial cells in RA lesions [66,67]. NF-κB is also known to increase the production of the proinflammatory cytokine TNFα, and it should be noted that anti- TNFα antibodies exhibit dramatic clinical effects in the treatment of RA.

## 6. Signaling Lymphocytic Activation Molecule-Associated Protein (*SAP*)/*SH2D1A* and RA

X-linked lymphoproliferative disease type 1 (XLP-1) is a primary immunodeficiency characterized by aberrant immune responses following primary EBV infection. Impaired control of EBV infection often leads to fatal EBV-positive lymphoproliferation, hemophagocytic lymphohistiocytosis, and B-cell lymphomas. In 1998, the gene *SAP/SH2D1A* on the X chromosome was identified as being responsible for the development of XLP-1 [68]. *SAP/SH2D1A* encodes the intracellular adaptor protein SAP that is involved in signal transduction downstream of the SLAM family of cell surface receptors [69,70]. *SAP/SH2D1A* is mainly expressed in T cells and NK cells and the deficiency of *SAP/SH2D1A* function is thought to impair the immunological control of EBV-infected B cells by these lymphocytes [71,72,73,74,75,76,77,78,79,80,81,82,83,84]. Before its identification as the cause of XLP-1, we had independently cloned a full-length *SAP/SH2D1A* cDNA (GenBank: AB586694.1) from mRNAs isolated from peripheral leukocytes of a patient with IgA nephropathy, which is also known as EBV-related disease. Although the relationship between *SAP/SH2D1A* and IgA nephropathy is not clear, we determined the complete nucleotide sequence of *SAP/SH2D1A* cDNA (USA Patent No. 6828428 and several other international patents). Because *SAP/SH2D1A* deficiency was identified as the cause of aberrant T-cell control of EBV in XLP-1 and EBV-specific T-cell activity was known to be reduced in patients with RA, we quantified the level of *SAP/SH2D1A* mRNA in the peripheral blood T cells of patients with RA. The results indicated that *SAP/SH2D1A* mRNA levels are significantly lower in RA patients than in controls, suggesting that the reduced expression of the gene may have some role in the impaired EBV-specific T-cell responses in RA. We cloned *SAP/SH2D1A* cDNAs from peripheral leukocytes of five patients with RA and determined their sequence; however, no mutation was found [85].

## 7. Can EBV Directly Cause RA?

The evidence described above that suggests the involvement of EBV in RA pathogenesis, including elevated EBV-specific antibody levels [45,46] and EBV DNA load in patients [13], reduced EBV-specific T-cell responses in patients [63], molecular mimicry of RA autoantigens by EBV [46,49,50], and the presence of EBV and its products in RA synovial lesions [56,57,58,59], is all indirect evidence; no clear causal relationship between EBV and RA has been obtained. One of the obvious reasons for this delay in research has been the lack of suitable animal models of EBV infection. Humans are the only natural host of EBV infections and only limited species of laboratory animals can be experimentally infected with EBV. This difficulty has been at least partially overcome by the development of humanized mice. Transplantation of human hematopoietic stem cells to mice of severely immunodeficient strains including NOD-SCID (NOD/Shi-scid/IL-2Rγnull mice) (NOG) has been shown to result in the reconstitution of major components of the human immune system, including T, B, NK lymphocytes, macrophages, and dendritic cells [86,87,88]. As human B cells, a major cellular target of EBV infection, and human T cells, playing a central role in the immune response to the virus, are reconstituted in humanized mice, several groups have succeeded in recapitulating key features of human EBV infection in humanized mice [87,89,90,91]. We showed that EBV infection of humanized mice induced erosive arthritis histologically similar to RA in the knee joint. This arthritis is characterized by synovial cell proliferation and bone destruction accompanied by the unique “pannus” structure, a histological hallmark of RA. The bone marrow near the affected joint exhibited CD4+ T-cell infiltration and signs of bone marrow edema, known as an early RA manifestation in humans [92]. The pannus structure generated in this mouse model contained human osteoclasts [93]. Furthermore, we demonstrated that RANKL, an essential osteoclast-inducing cytokine, is produced by EBV-infected B cells [Iwata M, Nagasawa Y, Kitamura N, et al. Arthritis Rheumatol.2016;68(suppl 10) abstract No.471]. It is thus suggested that interaction between EBV-induced RANKL on the surface of infected B cells and RANK on osteoclast progenitors leads to the differentiation of osteoclasts, which have a role in bone erosion in RA.

## 8. Conclusions

In our previous articles on viruses and autoimmunity, the association between EBV and RA has been described in reference to the findings from our own laboratory [91,92,93,94,95,96,97,98,99,100]. In this paper, we have included the latest knowledge as much as possible.

Treatment of acute herpesvirus infections has been established using antiviral drugs and immunoglobulins. However, there is no curative therapy for EBV infection. There have been reports of vaccine development against EBV [101] and adoptive immunotherapy in which cytotoxic T cells were induced in vitro and then returned to patients [102]. Analysis of the mechanism of humoral immune memory originating from the discovery of the XLP-1-causing gene *SH2D1A* might lead to the development of new treatments of autoimmune diseases related to this virus.

We have been investigating the causal relationship between EBV and RA since the time when the presence of the virus in RA synovitis lesions was not yet known. During the course of our investigation, we cloned the *SH2D1A* gene that is critically involved in immune protection against EBV and applied humanized mice to generate a mouse model of EBV-induced RA-like arthritis (Figure 1). As reported by Tsumiyama et al. (self-organized criticality theory) [103], an autoimmune phenomenon is not always essential for the onset of autoimmune diseases. We sincerely hope that the knowledge accumulated so far will be useful in developing new treatments for rheumatic diseases.

## Figures and Tables

**Figure 1 viruses-14-01900-f001:**
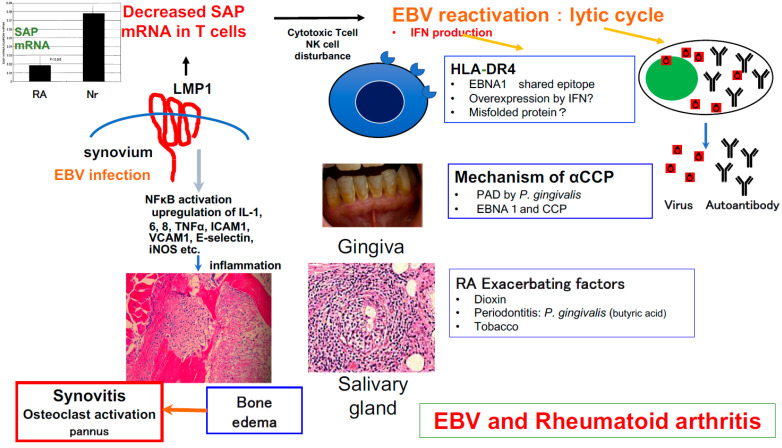
Inferring the role of EBV in RA synovitis. EBV is present in RA synovium, and LMP-1 expressed in synovial lining cells may suppress SAP function [56,104] and breaks down the normal defense mechanism against infection with EBV [85,103]. EBV activation might contribute to synovial cell proliferation, inflammation, and autoantibody production [52], and may be involved in the induction of RA synovitis. Additionally, environmental factors may contribute to EBV reactivation as follows: (1) *Porphyromonas gingivalis* peptidylarginine deiminase (PAD), an enzyme required for citrullination, engenders antigens leading to production of citrullinated peptides both in the gingiva and synovium [105]. (2) This bacterium also produces butyric acid, which induces EBV reactivation [106]. (3) Enhanced lytic replication of EBV induced by dioxin and tobacco is also a risk factor for the development of autoimmune diseases [107,108]. EBV infection in humanized NOG mice resulted in a pathological condition characterized by erosive arthritis that is also seen in human rheumatoid arthritis; EBV infection is thus thought to be directly involved in RA pathogenesis [92].

**Table 1 viruses-14-01900-t001:** Association of viral infections with autoimmune diseases.

Virus	Autoimmune Disease	Reference
Parvovirus B19	SLE, adult-onset Still’s disease, RA	[8]
Epstein–Barr virus	RA, SLE, SS, PN, IgA nephropathy, MS, autoimmune thyroiditis, G-B syndrome, ITP	[5,6,7,9,10,11,12,13,14,15,16,36]
Human cytomegalovirus	SLE, PN, G-B syndrome	[17,37]
Varicella-zoster virus	ITP, SLE, RA, inflammatory bowel disease, autoimmune hepatitis	[15,16,18,19]
Influenza virus	ITP, type I DM	[15,20]
Rubella virus	ITP, type I DM	[15,21]
Human T-cell lymphotropic virus type 1	HAM, SS, RA, PM, SSc, SLE	[22]
Human immunodeficiency virus type 1	SS, SLE, APS, PBC, PM, autoimmune hepatitis, vasculitis, ITP	[23,24]
Hepatitis B virus	PN, cryoglobulin vasculitis	[25]
Hepatitis C virus	SS, RA, PN, cryoglobulin vasculitis	[17,38]
Coxsackie B virus	Type I DM	[17,26]
Rotavirus	Type I DM	[16,27]

SLE: systemic lupus erythematosus, RA: rheumatoid arthritis, SS: Sjogren disease, PN: polyarteritis nodosa, G-B syndrome: Guillain–Barré syndrome, ITP: idiopathic thrombocytopenia, DM: diabetes mellitus, HAM: HTLV-1 associated myelopathy, PM: polymyositis, SSc: systemic sclerosis, APS: antiphospholipid syndrome, PBC: primary biliary cholangitis.

**Table 2 viruses-14-01900-t002:** Common manifestations of rheumatic diseases caused by viral infections.

Frequency	Manifestation		
	Arthritis [28,29]	Uveitis [30,31,32,33]	Erythema Nodosum [34]
Often	HBV, HCV, rubella (adult female), parvovirus B19, HTLV-1, HIV	Herpesvirus family (HSV, HCMV, VZV) in AIDS	
Sometimes	Macacine alphaherpesvirus 1, Mayaro virus, mumps virus, VZV	HTLV-1, rubella virus	HBV, EBV, HIV, cytomegalovirus
Rare	Adenovirus type 7, herpesvirus family (EBV, HCMV, VZV)	Filovirus (Ebola), arbovirus (West Nile)	

Confer [35].

**Table 3 viruses-14-01900-t003:** Autoimmune processes with possible roles of viral infections.

Polyclonal B cell activation
Dual TCR-expressing T cells
Molecular mimicry
Antigen spreading
Bystander activation
HLA dysfunction
Regulatory T cell dysfunction
Aberrant function of signal transduction; SAP etc.

Pathogenic cytokine secretion from virus-infected cells -especially Interferon-. Misfolded proteins complexed with HLA, Attenuation of regulatory T cell function etc.

## Data Availability

No new data were created or analyzed in this study. Data sharing is not applicable to this article.

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
