# Peer review of "Are Viral Infections Key Inducers of Autoimmune Diseases? Focus on Epstein–Barr Virus"

_viruses, 2022, doi:10.3390/v14091900_

Round 1

Reviewer 1 Report

This is a good review. However, Tables are poorly organised and have no references. They must be improved to be systematic and easily readable.

Refernces in the text are sparse in many places. Referrences appear as x) and this makes them hard to read.

Several recent reviews on EBV and rheumatic diseases and multiple sclerosis should have been mentioned.

Minor:

Line:

81 into the blood stream

88 The immune

89 neo-self has been reported

133 EBNA (EBV nuclear antigen)

171 cells. (delete "proliferation")

188 Meanwhile, 

234 One of the obvious

282 Insert space after period

Author Response

Thank you very much for your helpful and constructive comments. We have revised the manuscript including the tables according to your comments.

This is a good review. However, Tables are poorly organised and have no references. They must be improved to be systematic and easily readable.

Response: We have reorganized Table 1 and Table 2 and included references. 

References in the text are sparse in many places. References appear as x) and this makes them hard to read.

Response: We have added some references to the text and changed the style of quotation in accordance with that of MDPI.

Several recent reviews on EBV and rheumatic diseases and multiple sclerosis should have been mentioned.

Response: We have added new sentences into the Introduction section and included new references (line - ).

Minor:

Response: Thank you very much for pointing out these errors in English. We have corrected them according to your suggestions.

Line:

81 into the blood stream

88 The immune

89 neo-self has been reported

133 EBNA (EBV nuclear antigen)

171 cells. (delete "proliferation")

188 Meanwhile,

234 One of the obvious

282 Insert space after period

Reviewer 2 Report

In this manuscript, Takei et al reviewed the potential involvments of viruses in the pathogenesis of autoimmune diseases, with a special emphasis on EBV. The paper is clearly written, interesting to read and timely.

Here are some concerns :

Line 52 : « Typically, parvo virus B19, hepatitis virus B and C, human T-cell leukemia virus type 1, rubella virus, HIV, and CMV are known to induce autoimmune diseases (Table 1) ». To be reformulated and weighted. To date, there is no scientific evidence to confirm that these viruses induce autoimmune diseases, but rather a bundle of presumption.

Table 1: references should be added, and the formatting reworked

Line 92 : Remove "Recently"

Line 148 : Authors should explain in two or three sentences how the fact that the gp110 contains a QKRAA sequence shared by HLA-DR4 and -1 could lead to the onset of RA.

Typos :

Line 53 : parvovirus B19 , not parvo virus B19

Line 268 

Author Response

In this manuscript, Takei et al reviewed the potential involvements of viruses in the pathogenesis of autoimmune diseases, with a special emphasis on EBV. The paper is clearly written, interesting to read and timely.

Here are some concerns :

Response: Thank you very much for your valuable comments and we have revised the manuscript in accordance with them.

Line 52 : « Typically, parvovirus B19, hepatitis virus B and C, human T-cell leukemia virus type 1, rubella virus, HIV, and CMV are known to induce autoimmune diseases (Table 1) ». To be reformulated and weighted. To date, there is no scientific evidence to confirm that these viruses induce autoimmune diseases, but rather a bundle of presumption.

Response: I agree with this comment and modified the sentence to indicate that these are hypotheses rather than strong evidence. The revised sentence reads “Typically, parvovirus B19, hepatitis virus B and C, human T-cell leukemia virus type 1, rubella virus, HIV, and CMV have been implicated in the development of are known to induce autoimmune diseases (Table 1).”

Table 1: references should be added, and the formatting reworked

Response: We have reorganized the table and added references.

Line 92 : Remove "Recently"

Response: We removed “Recently”.

Line 148 : Authors should explain in two or three sentences how the fact that the gp110 contains a QKRAA sequence shared by HLA-DR4 and -1 could lead to the onset of RA.

Response: We have added the following sentence in the revised manuscript (line - ). It reads “Both humoral and T-cell-mediated cross-reactivities between EBV gp110 and HLA-DR4 have been described, although the exact mechanism of their involvement in RA pathogenesis is not clear.”.

Typos :

Line 53 : parvovirus B19 , not parvo virus B19

Response: We have corrected the word.

Line 268

Response: We have modified the sentence that starts at line 268 as follows; We have been investigating the causal relationship between EBV and RA since the time when the presence of the virus in RA synovitis lesions was not yet known. Is this what you meant?

Reviewer 3 Report

Masami TAKEI et al. prepared a manuscript entitled “Are viral infections key inducers of autoimmune diseases? 2 -Focus on Epstein-Barr virus.

The concept of the thesis itself is good, but it requires supplementation and a large technical correction, in particular regarding the quotation of the literature.

Major revisions:

1. Introduction - needs to be completed. Expand the introduction to data on microRNA and Viruses and autoimmune diseases. Only publication no [5] was given. 

Minor revisions:

1. Manuscript prepared very carelessly.

2. Please provide references in the appropriate form for examples [2,3].

3. Table 1 is illegible, please prepare it informatively, not in cipher. Prepare the table in accordance with the MDPI style.

4. Evil citing references:

Line 92 -Recently, Qingyong and others proposed a new mechanism of the virus-induced devel- 92 opment of autoimmune disease. They identified CD8 + T cells with dual T cell receptors 93 that recognize a self-antigen and a virus-specific protein 12).

It should be: Recently, Qingyong and others proposed a new mechanism for the virus-induced devel- 92 opment of autoimmune disease. They identified CD8 + T cells with dual T cell receptors 93 that recognize a self-antigen and a virus-specific protein 12).

5. Insert the Tables in the appropriate place in the text, after the first reference to the Table.

Standardize spelling throughout the work:

There is Qingyong and others (line 92); Tan et al. (line 144)

Should be: Qingyong et al. [X], Tan et al. [X], and so on ...

Line 144- Harley et al. [x]

Line 174- Saal et al. [x]

Line 177-Bad Quote. Please give this information a reference number in the literature "Iwata and Tateishi demonstrated the 177 presence of EBV in mononuclear cells of RA in the subchondral zone by in situ hybridization and immunohistochemical staining (The 43rd Annual Meeting of the Japanese 179 Viruses 2022, 14, x FOR PEER REVIEW 8 of 13 Association for Rheumatology F84-4, 1999) ”.

Line 180-Correct - Mahraein et al. on Mehraein [2] and write parenthesis [29].

29. Mehraein Y. Lennerz C, Ehlhardt S, et al. : Latent Epstein-Barr virus (EBV) infection and cytomegalovirus (CMV) infection in 355 synovial tissue of autoimmune chronic arthritis determined by RNA- and DNA-in situ hybridization. Mod Pathol. 2004; 17: 781-356 9.

Line 180-203 -correct references citation.

Line 230-242-Provide references on the facts, not summarized.

Line 251-256-Lack references.

Line 257-275-Correct references- Shiozawa et al. [74].

Author Response

Masami TAKEI et al. prepared a manuscript entitled “Are viral infections key inducers of autoimmune diseases? 2 -Focus on Epstein-Barr virus. The concept of the thesis itself is good, but it requires supplementation and a large technical correction, in particular regarding the quotation of the literature.

Response: Thank you very much for your valuable suggestions. We have reorganized both Tables 1 and 2 and added some new sentences to the text.  The style of quotation of references has been changed to what is compatible with MDPI. 

Major revisions:

  1. Introduction - needs to be completed. Expand the introduction to data on microRNA and Viruses and autoimmune diseases. Only publication no [5] was given.

Response: We have added a sentence to the Introduction section; Although the precise mechanisms of virus-induced autoimmunity have not been elucidated, recent evidence suggests that microRNA could play important roles [reviewed in a2, a3].

Minor revisions:

  1. Manuscript prepared very carelessly.

Response: We apologize for the errors in style and language.  We have corrected them as precisely as possible.

  1. Please provide references in the appropriate form for examples [2,3].

Response: We have corrected the style of quotation of references throughout the manuscript.

  1. Table 1 is illegible, please prepare it informatively, not in cipher. Prepare the table in accordance with the MDPI style.

Response: We have reorganized Table 1 and added references to it.

  1. Evil citing references:

Line 92 -Recently, Qingyong and others proposed a new mechanism of the virus-induced devel- 92 opment of autoimmune disease. They identified CD8 + T cells with dual T cell receptors 93 that recognize a self-antigen and a virus-specific protein 12).

It should be: Recently, Qingyong and others proposed a new mechanism for the virus-induced devel- 92 opment of autoimmune disease. They identified CD8 + T cells with dual T cell receptors 93 that recognize a self-antigen and a virus-specific protein 12).

Response: Thank you for this helpful suggestion. We have corrected the sentence exactly as suggested.

  1. Insert the Tables in the appropriate place in the text, after the first reference to the Table.

Response: Thank you for this helpful suggestion. We have Inserted the Tables in the appropriate place in the text as suggested.

Standardize spelling throughout the work:

There is Qingyong and others (line 92); Tan et al. (line 144)

Should be: Qingyong et al. [X], Tan et al. [X], and so on ...

Response: We have corrected this inconsistence.

Line 144- Harley et al. [x]

Line 174- Saal et al. [x]

Line 177-Bad Quote. Please give this information a reference number in the literature "Iwata and Tateishi demonstrated the 177 presence of EBV in mononuclear cells of RA in the subchondral zone by in situ hybridization and immunohistochemical staining (The 43rd Annual Meeting of the Japanese 179 Viruses 2022, 14, x FOR PEER REVIEW 8 of 13 Association for Rheumatology F84-4, 1999) ”.

Response: We have deleted this sentence.

Line 180-Correct - Mahraein et al. on Mehraein [2] and write parenthesis [29].

  1. Mehraein Y. Lennerz C, Ehlhardt S, et al. : Latent Epstein-Barr virus (EBV) infection and cytomegalovirus (CMV) infection in 355 synovial tissue of autoimmune chronic arthritis determined by RNA- and DNA-in situ hybridization. Mod Pathol. 2004; 17: 781-356 9.

Response: We have corrected the style of quotation.

Line 180-203 -correct references citation.

Response: We have corrected the style of quotation.

Line 230-242-Provide references on the facts, not summarized.

  Response: We have added references to individual findings suggesting involvement of EBV in RA.

Line 251-256-Lack references.

 Response: We have corrected the reference [Arthritis Rheumatol. 2016; 68 (suppl 10) ]

Line 257-275-Correct references- Shiozawa et al. [74].

Response: We have corrected the reference (Tsumiyama et al [74]).

Round 2

Reviewer 1 Report

The manuscript has been improved and is suitable for publication. The use of additional references for tables is a little confusing, so it should be mentioned in table 1 that the references for the table are found after the reference list.

Author Response

Thank you very much for your helpful and kind review. We have revised the manuscript including the tables according to your comments. We reorganized Table 1 and added some references to clarify. And also we have changed the formatting of the references.
